# High-Probability Bounds for Robust Stochastic Frank-Wolfe Algorithm

Tongyi Tang[1]          Krishnakumar Balasubramanian[1]          Thomas C. M. Lee[1]

[1]Department of Statistics, University of California, Davis, CA, USA.

## Abstract

We develop and analyze robust Stochastic Frank-Wolfe type algorithms for projection-free stochastic convex optimization problems with heavy-tailed stochastic gradients. Existing works on the oracle complexity of such algorithms require a uniformly bounded variance assumption, and hold only in expectation. We develop tight high-probability bounds for robust versions of Stochastic Frank-Wolfe type algorithm under heavy-tailed assumptions, including infinite variance, on the stochastic gradient. Our methodological construction of the robust Stochastic Frank-Wolfe type algorithms leverage techniques from the robust statistic literature. Our theoretical analysis highlights the need to utilize robust versions of Stochastic Frank-Wolfe type algorithm for dealing with heavy-tailed data arising in practice.

## 1 INTRODUCTION

In this paper, we consider constrained stochastic optimization problem of the form

$$\arg\min_{x \in \mathcal{X}} \ \{f(x) = \mathbb{E}_\xi[F(x,\xi)] = \int F(x,\xi)\,d\mathbb{P}(\xi)\}, \ (1)$$

where $f$ is a smooth convex function, $\mathcal{X}$ is a closed convex subset of $\mathbb{R}^d$, $\xi$ is a random vector defined according to a distribution $\mathbb{P}$ on the domain of $\xi$. We denote by $x^*$, the minimizer of $f(x)$ in (1). Problems of the form in (1) arise frequently in modern machine learning, including matrix completion [Davenport and Romberg, 2016, Freund et al., 2017], structured linear inverse problems [Chandrasekaran et al., 2012, Tewari et al., 2011], multi-task learning [Sener and Koltun, 2018, Zhao et al., 2020] and particle filtering [Lacoste-Julien et al., 2015]. Stochastic Frank-Wolfe (SFW) and Stochastic Conditional Gradient Sliding

(SCGS) algorithms are widely used iterative first-order algorithm for solving (1). Each iteration of SFW/SCGS involves performing linear minimization over the constraint set $\mathcal{X}$ based on stochastic first-order (or gradient) information, $\nabla F(\cdot, \xi) \in \mathbb{R}^d$. Depending on the geometry of the constraint set, SFW/SCGS is widely used in practice due to its projection-free nature [Besançon et al., 2021]. Despite its wide-spread usage, our understanding of the oracle complexity of SFW/SCGS algorithm is limited.

As each iteration of the SFW/SCGS algorithm requires access to a Stochastic First-order Oracle (SFO) and a Linear Minimization Oracle (LMO), the oracle complexity is measured by the number of calls to SFO and LMO to obtain an $\epsilon$-optimal solution – that is, a point $\bar{x}$ such that $f(\bar{x}) - f(x_*) \leq \epsilon$. A majority of the existing results on the oracle complexity of SFW/SCGS algorithm are available only in expectation, i.e., on the metric $\mathbb{E}_\xi[f(\bar{x}) - f(x_*)] \leq \epsilon$. Furthermore, to obtain such oracle complexity results in expectation, it is assumed that stochastic gradient used has uniformly bounded variance (i.e., $\mathbb{E}_\xi[\|\nabla F(x,\xi) - \nabla f(x)\|^2] \leq \sigma^2$), where $\|\cdot\|$ denotes the Euclidean norm on $\mathbb{R}^d$. This characterization of the oracle complexity provides very little understanding regarding the behaviour of the SFW/SCGS algorithms. In particular, the effect of the properties of the distribution $\mathbb{P}$, on the heavy-tail nature of the stochastic gradient, and consequently on the oracle complexity is *not characterized* by the oracle complexity results in expectation. Furthermore, the oracle complexity of SFW/SCGS algorithm or its robust variants under infinite variance assumption is not known.

To provide a concrete motivating example, consider the problem of sparse linear regression: Given $(a, y) \in \mathbb{R}^d \times \mathbb{R}$, consider the linear regression model $y = \langle a, \bar{x} \rangle + \epsilon$, where for two vectors $c, d \in \mathbb{R}^d$, $\langle c, d \rangle$ represents the Euclidean inner product. Here, we let $\epsilon \sim N(0, 1)$ and the true parameter $\bar{x}$ is assumed to be $s$-sparse (i.e., it has only $s$ non-zero coordinates out of the $d$ coordinates). The $L_1$-constrained

*Accepted for the 38th Conference on Uncertainty in Artificial Intelligence* (UAI 2022).

least-squares estimator is then given by

$$\underset{x \in \mathcal{X}_1(s)}{\arg\min} \ \mathbb{E}[(y - \langle a, x \rangle)^2], \qquad (2)$$

where $\mathcal{X}_1(s) := \{x \in \mathbb{R}^d : \sum_{j=1}^{d} |x_j| \le s\}$ is the $L_1$ ball of radius $s$. This problem fits in the setup of (1) with $\xi := (a, y)$ and $F(x, \xi) := (y - a^\top x)^2$. Hence the stochastic gradient is given by

$$G(x, \xi) := \nabla F(x, \xi) = 2(aa^\top x - ya) \in \mathbb{R}^d.$$

Note that as the iterates of SFW/SCGS algorithms (see, for example, Algorithm 2 for the description of SCGS) are in the set $\mathcal{X}_1(s)$, we have $\|x\|$ is to be always bounded for all $x$ along the trajectory of the algorithm. Hence, the $(1 + \alpha)$-th moment of the stochastic gradient, i.e., $\mathbb{E}[\|G(x, \xi)\|^{(1+\alpha)}]$, is controlled by the order of $\mathbb{E}[\|a\|^{2(1+\alpha)}]$. When the covariate $a$ is a zero-mean multivariate $t$-distribution with degrees of freedom in the interval $[2, 4]$, or is a zero-mean multivariate Pareto distribution with parameter in the interval $[2, 4]$, the stochastic gradients have infinite variance. Hence, the existing oracle complexity results for SFW/SCGS are not applicable.

Focusing on the SCGS algorithm [Lan and Zhou, 2016], in this work we develop and analyze robust versions of it, and establish high-probability SFO and LMO complexity bounds. First, under a sub-Gaussian tail assumption on the stochastic gradient, we establish high-probability bounds for the standard SCGS algorithm. However, when the stochastic gradient is heavy-tailed or has infinite variance, the standard SCGS algorithm is sub-optimal. This is due to the fact that each iteration of SCGS algorithm requires a sample average of mini-batch stochastic gradients. When the stochastic gradients are heavy-tailed, it is well-known from classical robust statistics literature that sample averages are poor estimates of the true expectation. Hence, to deal with the case when the stochastic gradient is heavy-tailed or has infinite variance, we construct robust versions of mini-batch stochastic gradient estimates which are subsequently used in the SCGS algorithm. Our motivations for developing robust SCGS algorithms are based on the fact that data arising from various real-world problems (for example, finance, networks, linguistics) are modeled efficiently using heavy-tailed distribution [Resnick, 2007, Fan et al., 2014, Taleb, 2020, Roughgarden, 2020]. We establish high-probability bounds on the oracle complexity results on robust SCGS algorithm by developing tight concentration inequalities for heavy-tailed martingales, which might be of independent interest. The established high-probability bounds on the SFO and LMO complexity of SCGS, provide a fine-grained characterization of the effect of the moments/tails of the distribution $\mathbb{P}$ on the performance of SCGS algorithms in terms of both the level of solution accuracy and the confidence specified by the practitioner.

## 1.1 RELATED WORKS

**Robust statistics:** Robust statistics is a classical topic with too large a literature to summarize completely. We refer the reader to Huber [2004] for an overview. We emphasize that a majority of the robust estimators developed in the statistics literature are invariably computationally intractable. The revival of robust statistics in modern mathematical statistics and learning theory communities arguably started with the work of Catoni [2012]. Since then, there has been intense work on robust mean and covariance estimation [Minsker, 2015, Cardot et al., 2017, Minsker, 2018, Lugosi and Mendelson, 2019b,a, Hopkins, 2020], and robust empirical risk minimization [Hsu and Sabato, 2016, Diakonikolas et al., 2019, Geoffrey et al., 2020, Lecué and Lerasle, 2020]. However, such results are mainly statistical in nature and are not directly applicable for stochastic optimization with heavy-tailed gradients. In this work, we leverage the classical trick of *trimmed estimator* from the robust statistics literature, and the recently proposed optimal mean-estimation technique by Cherapanamjeri et al. [2022] in the context of projection-free stochastic optimization.

**Robust stochastic optimization:** Early works on robust stochastic optimization include Krasulina [1969], Martin and Masreliez [1975], Price and VandeLinde [1979], Chen and Gao [1989], Nemirovskij and Yudin [1983]. Limit theorems for random iterated maps and time-series with heavy-tails were proved in Mirek [2011], Buraczewski et al. [2012], Mikosch and Wintenberger [2016]. Such results make restrictive assumptions that are not satisfied by stochastic optimization algorithms. In modern stochastic optimization, several works consider the effect of heavy-tails on the performance of the algorithm [Oliveira and Thompson, 2017, Nazin et al., 2019, Wang et al., 2021, Davis et al., 2021, Bartl and Mendelson, 2022, Anantharam and Borkar, 2012, Holland, 2021]. There has also been overwhelming theoretical and empirical evidence in the modern machine learning literature that shows the noise in stochastic gradient algorithm could easily turn out to have heavy-tails due to the composite or product nature of the random vectors/matrices in the stochastic gradient [Hodgkinson and Mahoney, 2021, Panigrahi et al., 2019, Simsekli et al., 2019, 2020b, Gurbuzbalaban and Hu, 2021, Camuto et al., 2021, Scaman and Malherbe, 2020, Simsekli et al., 2020a]. Furthermore, recently robust mean estimation based and trimming based SGD algorithms were analyzed in Prasad et al. [2020] and Bubeck et al. [2013], Gorbunov et al. [2020], Zhang et al. [2020b], Mai and Johansson [2021], Zhang et al. [2020a] respectively. However the above works do not deal with projection-free robust stochastic optimization, which is the main focus of our work.

**Frank-Wolfe Algorithms:** The Frank-Wolfe method first proposed by Frank and Wolfe [1956], Levitin and Polyak [1966], has had a renewed interest in the past decade. We refer the reader to Jaggi [2013], Harchaoui et al. [2015],

Lacoste-Julien and Jaggi [2015], Beck and Shtern [2017], Garber et al. [2018], for a partial list of recent works in the deterministic setting. Considering the stochastic convex setup, Hazan and Kale [2012], Hazan and Luo [2016] provided expected oracle complexity results for SFW algorithm. The complexities were further improved by a sliding procedure in Lan and Zhou [2016], based on a modified Frank-Wolfe method by Nesterov's acceleration. It is common in SFW/SCGS analysis to require an increasing batch-size in each step, to obtain oracle complexity results. Recently, Mokhtari et al. [2020], Hassani et al. [2020], Zhang et al. [2020c] addressed this issue of increasing batch size. But these works require restrictive assumptions, and have sub-optimal LMO complexity. In section 4 we empirically compare against the 1-sample SFW method from Mokhtari et al. [2020] in the heavy-tailed setting. We also highlight that several works [Reddi et al., 2016, Yurtsever et al., 2019, Hazan and Luo, 2016] considered variance reduced versions of SFW and provided expected oracle complexities. In comparison to the above works, we focus on high-probability bounds on the oracle complexity in both the light-tailed and heavy-tailed setting (including infinite variance).

## 1.2 OUR CONTRIBUTIONS

We now provide a list of theoretical contributions to the literature on oracle complexity of SFW algorithm. To do so, we first introduce the notion of optimality that we consider for solving (1).

**Definition 1.1** (($\epsilon, \delta$)-optimal solution). We call $\bar{x}$ an ($\epsilon, \delta$)-optimal solution of the stochastic optimization problem (1), if we have $\mathbb{P}\left(f(\bar{x}) - f(x_*) \leq \epsilon(\delta)\right) \geq 1 - \delta$. Here, the function $\epsilon(\delta)$ stands for the target accuracy and the parameter $\delta$ stands for the required level of confidence.

In contrast to the oracle complexity results obtained in the literature which are only in expectation, the above definition takes into account an user-specified level of confidence $\delta$, with which we could obtain an $\epsilon$-optimal solution of the stochastic optimization problem (1). This helps obtain a fine-grained characterization of the moments/tails of $\mathbb{P}$ on the oracle complexity of SFW algorithms. We next list the specific notion of light and heavy-tailed assumption that we make on the stochastic first-order oracle.

**Assumption 1.2** (Sub-Gaussian). For any $x \in \mathbb{R}^d$, the SFO outputs an estimator $G(x, \xi)$ of $\nabla f(x)$ such that $\mathbb{E}[G(x, \xi)] = \nabla f(x)$, and $\mathbb{E}[\exp\{\|G(x, \xi) - \nabla f(x)\|^2 / \sigma^2\}] \leq \exp\{1\}$, for some $\sigma^2 > 0$.

*Remark* 1.3. The sub-Gaussian tail assumption is satisfied by light-tailed distributions arising in practice, and hence has been used widely in statistics and stochastic optimization [Vershynin, 2018, Wainwright, 2019]. It should be note Assumption 1.2 implies by Jensen's inequality that $\mathbb{E}[\|G(x, \xi) - \nabla f(x)\|^2] \leq \sigma^2$, which is used to obtain or-

acle complexity results for SFW in expectation. However, the converse is not true.

**Assumption 1.4** (Weak-exponential). For any $x \in \mathbb{R}^d$, the first-order oracle outputs an estimator $G(x, \xi)$ of $\nabla f(x)$ such that $\mathbb{E}[G(x, \xi)] = \nabla f(x)$, and $\mathbb{E}[\|G(x, \xi)\|^{1+\alpha}] \leq \sigma^{1+\alpha}$ where $\alpha \in (0, 1]$.

**Assumption 1.5** (Weak-moment). For any $x \in \mathbb{R}^d$, the first-order oracle outputs an estimator $G(x, \xi)$ of $\nabla f(x)$ such that $\mathbb{E}[G(x, \xi)] = \nabla f(x)$, and for all $\|v\| = 1$, we have $\mathbb{E}[|\langle G(x, \xi) - \nabla f(x), v \rangle|^{1+\beta}] \leq 1$ for some $\beta \in (0, 1]$

*Remark* 1.6. Firstly, we point out that both Assumptions 1.4 and 1.5 allow for the stochastic gradient to have infinite variance (when $\alpha, \beta < 1$). Note that in Assumption 1.5 the vector $v$ is normalized to 1, so $\beta$ does not appear in the RHS. Without loss of generality, we set constant on right hand side to 1 as any finite positive constant anyway can will absorbed in the $\mathcal{O}$ notation. Furthermore, even when $\alpha, \beta = 1$, Assumptions 1.4 and 1.5 allow for tails heavier than the sub-Gaussian tail in Assumption 1.2. For canonical problems like structured linear or multi-class logistic regression, this assumption allows for the covariate to be heavy-tailed including Pareto or $t$-distribution, which do not satisfy Assumption 1.2. Furthermore, Assumption 1.4 is stronger than Assumption 1.5. Indeed if $\zeta \in \mathbb{R}^d$ is a random vector that satisfies Assumption 1.5, then we have $\mathbb{E}[\|\zeta\|^{1+\beta}] \leq (\pi/2) d^{\frac{1+\beta}{2}}$.

With the above preliminaries, we now present our **main contributions** in this work.

- We first characterize the SFO and LMO complexity, in high-probability, when the standard mini-batch average of the stochastic gradients is used, in Theorem 3.3 under the condition that the distribution of the stochastic gradients follows the sub-Gaussian tail Assumption 1.2.
- We next establish the SFO and LMO complexity, in high-probability, when using the clipped gradient estimators, under the heavy-tailed Assumptions 1.4 and 1.5 in Theorem 3.6 (a) and (b) respectively.
- Next, in Theorem 3.8, we show that the high-probability oracle complexity results under Assumption 1.5 could be further improved by using a gradient estimator based on a recently proposed optimal mean-estimator procedure in [Cherapanamjeri et al., 2022], at the cost of increased per-iteration complexity.
- Finally, we propose a computationally efficient biased-corrected clipped gradient procedure. We show in Theorem 3.13 that this approach obtains improved high-probability oracle complexities (over Theorem 3.6) results under an additional symmetry condition (see Assumption 3.10) on the distribution of the stochastic gradients, for certain regimes of $\epsilon$ and $d$.

A summary of our oracle complexity results is provided in Table 1 and visual illustrations are provided in Section 2.

| Mean-Estimator | Tails | Theorem | SFO | LMO |
|---|---|---|---|---|
| Average | Asmp. 1.2 | Thm. 3.3 | $\mathcal{O}\left(\left(\frac{\log(1/\delta)}{\epsilon}\right)^2\right)$ | $\mathcal{O}\left(\left(\frac{\log(1/\delta)}{\epsilon}\right)\right)$ |
| Clipped Grad. | Asmp. 1.4 | Thm. 3.6 (a) | $\mathcal{O}\left(\left(\frac{(\log(1/\delta))^{\frac{\alpha}{1+\alpha}}}{\epsilon}\right)^{\frac{3\alpha+2}{2\alpha}}\right)$ | $\mathcal{O}\left(\frac{(\log(1/\delta))^{\frac{\alpha}{1+\alpha}}}{\epsilon}\right)$ |
| Clipped Grad. | Asmp. 1.5 | Thm. 3.6 (b) | $\mathcal{O}\left(\left(\frac{\sqrt{d}(\log(1/\delta))^{\frac{\beta}{1+\beta}}}{\epsilon}\right)^{\frac{3\beta+2}{2\beta}}\right)$ | $\mathcal{O}\left(\frac{\sqrt{d}(\log(1/\delta))^{\frac{\beta}{1+\beta}}}{\epsilon}\right)$ |
| CTBJ [Cherapanamjeri et al., 2022] | Asmp. 1.5 | Thm. 3.8 | $\mathcal{O}\left(\left(\frac{(\log(1/\delta))^{\frac{\beta}{1+\beta}}}{\epsilon}\right)^{\frac{3\beta+2}{2\beta}}\right)$ | $\mathcal{O}\left(\frac{(\log(1/\delta))^{\frac{\beta}{1+\beta}}}{\epsilon}\right)$ |
| BC Clipped Grad. | Asmp. 1.4 | Thm. 3.13 (a) | $\mathcal{O}\left(\left(\frac{C(d,\alpha,\delta)}{\epsilon}\right)^{\frac{5\alpha+3}{4\alpha}}\right)$ | $\mathcal{O}\left(\frac{C(d,\alpha,\delta)}{\epsilon}\right)$ |
| BC Clipped Grad. | Asmp. 1.5 | Thm. 3.13 (b) | $\mathcal{O}\left(\left(\frac{C(d,\beta,\delta)}{\epsilon}\right)^{\frac{5\beta+3}{4\beta}}\right)$ | $\mathcal{O}\left(\frac{C(d,\beta,\delta)}{\epsilon}\right)$ |

Table 1: A summary of the obtained high-probability bounds. All results corresponds to the notion of $(\epsilon, \delta)$-optimal solution introduced in Definition 1.1. The parameters $C(d, \alpha, \delta)$ and $C(d, \beta, \delta)$ are defined in Theorem 3.13. BC stands for Bias-Corrected. The last two rows also require the symmetric condition described in Assumption 3.10.

To our knowledge, our work provides the *first comprehensive high-probability oracle complexity results* for SCGS algorithm with light and heavy-tailed stochastic gradients (including infinite variance). We discuss the consequences of our theoretical results for practice by reporting simulations for heavy-tailed sparse linear regression (Section 4) and multi-class logistic regression (Appendix–Section 1).

## 2 ROBUST STOCHASTIC FRANK-WOLFE ALGORITHMS

Recall that our task is to solve constrained stochastic convex optimization problems of the form in (1) using projection-free Frank-Wolfe type algorithm, which involves two main steps: the gradient evaluation step and the linear optimization step. The gradient evaluation step is typically based on averaging a mini-batch of gradients [Hazan and Luo, 2016, Lan and Zhou, 2016, Balasubramanian and Ghadimi, 2021]. It is well-known from classical robust statistics that the sample average is not an accurate estimate of true expectation in the heavy-tailed setting. Hence, a natural strategy is to replace the sample average with appropriate robust versions of mean estimators, and incorporate such estimators in Frank-Wolfe type algorithms. Specifically, we consider the version of stochastic conditional gradient sliding algorithm in Lan and Zhou [2016], also analyzed in Balasubramanian and Ghadimi [2021] for the case of biased gradients. We first state a subroutine in Algorithm 1 that we subsequently use.

Note that Algorithm 1 is indeed the SFW algorithm for inexactly solving the following quadratic program

$$P_{\mathcal{X}}(x, g, \gamma) = \arg\min_{u \in \mathcal{X}}\left\{\langle g, u\rangle + \frac{\gamma}{2}\|u - x\|^2\right\}, \quad (3)$$

which is the standard subproblem of stochastic first-order methods applied to a minimization problem when $g$ is an

---

**Algorithm 1** Inexact Conditional Gradient (ICG) method

Input: $(x, g, \gamma, \mu)$.
Set $\bar{y}_0 = x$, $t = 1$, and $\kappa = 0$..
**while** $\kappa = 0$ **do**
$\quad y_t = \arg\min_{u \in \mathcal{X}}\{h_\gamma(u) := \langle g + \gamma(\bar{y}_{t-1} - x), u - \bar{y}_{t-1}\rangle\}$
$\quad$ **If** $h_\gamma(y_t) \geq -\mu$, set $\kappa = 1$. **Else** $\bar{y}_t = \frac{t-1}{t+1}\bar{y}_{t-1} + \frac{2}{t+1}y_t$ and $t = t + 1$.
**end while**
Output $\bar{y}_{t-1}$.

---

**Algorithm 2** Robust Stochastic Accelerated Gradient Method with Inexact Updates

**Input:** $z_0 = x_0 \in \mathcal{X}$, positive integer sequence $m_k$, and sequences $\alpha_k$, $\gamma_k$, $\mu_k$ and iteration limit $N \geq 1$.
**for** $k = 1, \ldots, N$ **do**
$\quad$ 1. Set $w_k = (1 - \alpha_k)z_{k-1} + \alpha_k x_{k-1}$
$\quad$ 2. Call the stochastic oracle $m_k$ times, and compute (robust) stochastic gradients $\bar{G}_k$ as given by (4), (5), (6) or (10).
$\quad$ 3. Set $x_k = ICG(x_{k-1}, \bar{G}_k, \gamma_k, \mu_k)$, where $ICG(\cdot)$ is the output of Algorithm 1 with input $(x_{k-1}, \bar{G}_k, \gamma_k, \mu_k)$.
$\quad$ 4. Set $z_k = (1 - \alpha_k)z_{k-1} + \alpha_k x_k$.
**end for**
**Output:** $z_N$

---

unbiased stochastic gradient of the objective function at $x$. We now present Algorithm 2 which applies the Frank-Wolfe method to inexactly solve subproblems of the stochastic accelerated gradient method. It is well known that the above approach can significantly reduce the total number of calls to the stochastic oracle [Lan and Zhou, 2016, Balasubramanian and Ghadimi, 2021].

## 2.1 ROBUST MEAN ESTIMATORS

Note that step 2 in Algorithm 2 requires the gradient estimator denoted by $\bar{G}_k$. We now elaborate the robust stochastic gradient estimators that we propose to use in this work. Recall that standard analysis of SCGS algorithm takes the sample average of the mini-batch of i.i.d. stochastic gradient (obtained by querying the SFO) in each iteration. In this case, the gradient estimator is given by

$$\bar{G}_k := \frac{1}{m_k} \sum_{j=1}^{m_k} G(w_k, \xi_{k,j}). \tag{4}$$

As we will see from our analysis, the above gradient estimator is not robust to heavy-tails, i.e., when the vectors $G(w_k, \xi_{k,j})$ are heavy-tailed. Our first robust stochastic gradient estimator is based on the idea of trimmed or clipped estimators, generalized to the multivariate setting [Tukey and McLaughlin, 1963, Bickel, 1965, Huber, 2004, Stigler, 1973]. More recently such ideas have been used in the context of bandit optimization [Bubeck et al., 2013] and stochastic gradient descent algorithm [Gorbunov et al., 2020, Zhang et al., 2020a,b, Mai and Johansson, 2021]. Formally, in our setting, given i.i.d. stochastic gradients $\{G(w_k, \xi_{k,j})\}_{j=1}^{m_k}$, the *clipped gradient* estimator is defined, for some $\delta \in (0, 1)$,

$$\bar{G}_k := \frac{1}{m_k} \sum_{j=1}^{m_k} [G(w_k, \xi_{k,j}) \, \mathbb{1}\{A_j\}] \tag{5}$$

$$\text{with} \quad A_j := \|G(w_k, \xi_{k,j})\| \le \left( \frac{j\sigma^{1+\alpha}}{\log(1/\delta)} \right)^{\frac{1}{1+\alpha}}.$$

Here, $\mathbb{1}\{A\}$, for the event $A$ is defined as taking value 1 when the event A is true and taking value 0 otherwise. While the above estimator handles robust stochastic gradients (including ones with potential infinite variance condition in Assumption 1.4), it turns out that the oracle complexities under the even weaker condition in Assumption 1.5 with the above clipped gradient estimator is sub-optimal. To improve the oracle complexity under Assumption 1.5, we leverage the recent optimal robust mean estimation procedure proposed in Cherapanamjeri et al. [2022], which we call as the CTBJ procedure. Given i.i.d. stochastic gradients $\{G(w_k, \xi_{k,j})\}_{j=1}^{m_k}$, we use CTBJ estimator is given by (see Algorithm 1)

$$\bar{G}_k := \text{OPTIMALMEANEST}\left(\{G(w_k, \xi_{k,j})\}_{j=1}^{m_k}\right). \tag{6}$$

Roughly, the idea of Cherapanamjeri et al. [2022] is to use filtering to remove outliers and then compute the median by gradient descent procedures. A full description of the procedure is provided in Appendix–Section 5 for the sake of completeness. We also emphasize that while the estimator for the light-tailed case in (4) is unbiased, the robust gradient estimators in (5) and (6) are biased. This is another challenge that we handle in our analysis. Finally, we also remark that in Section 3.4, we introduce a bias-corrected clipped gradient procedure which achieves improved oracle complexities under an additional symmetry assumption on the distribution on stochastic gradients.

## 3 HIGH-PROBABILITY BOUNDS

We now provide our main results on high-probability bounds on the oracle complexity of Algorithm 2. To do so, we also make the following standard smoothness assumption about the stochastic gradient, which is common in the literature of smooth convex optimization [Nesterov, 2018, Lan and Zhou, 2016, Balasubramanian and Ghadimi, 2021].

**Assumption 3.1.** The objective function $f$ has Lipschitz continuous gradient with constant $L > 0$, i.e., $\|\nabla f(y) - \nabla f(x)\| \le L\|y - x\|$ for all $x, y \in \mathbb{R}^d$.

We first state a preliminary result about the iterates of Algorithm 2, under Assumption 3.1.

**Lemma 3.2.** *Let $\{z_k\}_{k \ge 1}$ be generated by Algorithm 2 and the function $f$ be convex. Let $\bar{\Delta}_k := \bar{G}_k - \nabla f(w_k)$, $\hat{\Gamma}_k := \prod_{i=2}^k (1 - \alpha_i)$, $\hat{\Gamma}_1 := 1$ and $D_0 = \|x_0 - x_*\|^2$. Then under Assumption 3.1, we have*

$$\frac{f(z_N) - f(x_*)}{\hat{\Gamma}_N} \le \frac{\gamma_1}{2}\|x_0 - x_*\|^2 + \sum_{i=1}^N \frac{\alpha_k \mu_k}{\hat{\Gamma}_k}$$
$$+ \sum_{i=1}^N \frac{\alpha_k}{\hat{\Gamma}_k} \langle \bar{\Delta}_k, x_* - x_{k-1} \rangle + \sum_{k=1}^N \frac{\|\bar{\Delta}_k\|^2}{2L\hat{\Gamma}_k}. \tag{7}$$

*For our subsequent analysis, we set*

$$\alpha_k = \frac{2}{k+1}, \quad \gamma_k = \frac{4L}{k}, \quad \text{and} \quad \mu_k = \frac{LD_0}{kN}. \tag{8}$$

The proof of Lemma 3.2 is provided in Appendix–Section 4 and is an intermediate result in the proof of Theorem 3.1 in Balasubramanian and Ghadimi [2021] with minor change. Our high-probability bounds are now based on developing concentration inequalities for the various gradient estimators $\bar{G}_k$ and bounding the terms appearing in right hand side of (7) in high-probability. To do so, we prove novel user-friendly concentration inequalities for (scalar-valued) martingales with heavy-tails that are discussed in detail in Section 3. It is worth mentioning that Lesigne and Volnỳ [2001] and Fan et al. [2017] also consider tail bounds for heavy-tailed martingales. However, they only provide deviation inequalities and their assumptions do no cover the regimes of heavy-tails that we are interested in.

## 3.1 ORACLE COMPLEXITY WITH SAMPLE AVERAGE ESTIMATOR

We now provide oracle complexity results for Algorithm 2 with the sample average gradient estimator in (4), that hold in high-probability.

**Theorem 3.3.** *Let Algorithm 2 be run with $\bar{G}_k$ as in (4), and with parameters $\alpha_k, \gamma_k$ and $\mu_k$ as in (8). If the stochastic gradients $G(x, \xi)$ satisfy Assumption 1.2 and 3.1, and $m_k = \mathcal{O}(N^3)$, then*

$$\mathbb{P}\left(f(z_N) - f(x_*) \leq \frac{D_0 \sigma^2 \log(1/\delta)}{N(N+1)}\right) \geq 1 - \delta,$$

*and the SFO and LMO complexity are respectively bounded by*

$$\mathcal{O}\left(\left(\frac{\log(1/\delta)}{\epsilon}\right)^2\right) \quad and \quad \mathcal{O}\left(\frac{\log(1/\delta)}{\epsilon}\right).$$

*Remark* 3.4. Note that [Lan and Zhou, 2016] shows that the SFO and LMO oracle complexity for Algorithm 2 with the sample average gradient estimator are of order $\mathcal{O}(1/\epsilon^2)$ and $\mathcal{O}(1/\epsilon)$ in expectation. Our results in Theorem 3.3 generalize this to the high-probability setting quantifying the effect of the allowed confidence level $\delta$ precisely.

## 3.2 ORACLE COMPLEXITY WITH CLIPPED GRADIENT ESTIMATOR

We first provide results on the bias, tail and moment bounds on the clipped gradient estimator $\bar{G}_k$ defined in (5). Then, we provide oracle complexity results for Algorithm 2 with the clipped gradient estimator, that hold with high-probability.

**Lemma 3.5.** *Let $\delta \in (0, 1)$ and $C$ be a positive universal constant. Let $\bar{G}_k$ be as defined in (5).*

*(a) If the stochastic gradients $G(x, \xi)$ satisfy Assumption 1.4, then we have*

$$\left\|\mathbb{E}[\bar{G}_k] - \nabla f(w_k)\right\| \leq \sigma \left(\frac{\log(1/\delta)}{m_k}\right)^{\frac{\alpha}{1+\alpha}}$$

$$and \quad \mathbb{P}\left(\|\bar{\Delta}_k\| \geq 4\sigma \left(\frac{\log(1/\delta)}{m_k}\right)^{\frac{\alpha}{1+\alpha}}\right) \leq \delta.$$

*Consequently, we also have the following moment bound*

$$\mathbb{E}\left[\exp\left\{\left\|\frac{\bar{\Delta}_k}{\sigma}\right\|^{\frac{1+\alpha}{\alpha}} m_k\right\}\right] \leq C.$$

*(b) If the stochastic gradients $G(x, \xi)$ satisfy Assumption 1.5, then we have*

$$\left\|\mathbb{E}[\bar{G}_k] - \nabla f(w_k)\right\| \leq \sqrt{d}\left(\frac{\log(1/\delta)}{m_k}\right)^{\frac{\beta}{1+\beta}}$$

$$and \quad \mathbb{P}\left(\|\bar{\Delta}_k\| \geq 4\sqrt{d}\left(\frac{\log(1/\delta)}{m_k}\right)^{\frac{\beta}{1+\beta}}\right) \leq \delta.$$

*Consequently, we also have the following moment bound*

$$\mathbb{E}\left[\exp\left\{\left\|\frac{\bar{\Delta}_k}{\sqrt{d}}\right\|^{\frac{1+\beta}{\beta}} m_k\right\}\right] \leq C.$$

**Theorem 3.6.** *Let Algorithm 2 be run with $\bar{G}_k$ as defined in (5), and with parameters $\alpha_k, \gamma_k$ and $\mu_k$ as defined in (8).*

*(a) Define $a(\delta, \omega) = (\log(1/\omega))^{\frac{\omega}{1+\omega}}$ and hence $a(\delta, \alpha) = (\log(1/\delta))^{\frac{\alpha}{1+\alpha}}$. If the stochastic gradients $G(x, \xi)$ satisfy Assumptions 1.4 and 3.1 and $m_k = \mathcal{O}(N^{\frac{2(\alpha+1)}{\alpha}})$, then*

$$\mathbb{P}\left(f(z_N) - f(x_*) \leq \frac{D_0 \sigma \max\left\{a(\delta, \alpha), \frac{\sigma}{N} a(\delta, \alpha)^2\right\}}{N(N+1)}\right)$$
$$\geq 1 - \delta,$$

*and the SFO and LMO complexity are respectively bounded by*

$$\mathcal{O}\left(\left(\frac{a(\delta, \alpha)}{\epsilon}\right)^{\frac{3\alpha+2}{2\alpha}}\right) \quad and \quad \mathcal{O}\left(\frac{a(\delta, \alpha)}{\epsilon}\right).$$

*(b) If the stochastic gradients $G(x, \xi)$ satisfy Assumptions 1.5 and 3.1 and $m_k = \mathcal{O}(N^{\frac{2(\beta+1)}{\beta}})$, then*

$$\mathbb{P}\left(f(z_N) - f(x_*) \leq \frac{D_0 \sqrt{d} \max\left\{a(\delta, \beta), \frac{\sqrt{d}}{N} a(\delta, \beta)^2\right\}}{N(N+1)}\right)$$
$$\geq 1 - \delta,$$

*and the SFO and LMO complexity are, respectively, bounded by*

$$\mathcal{O}\left(\left(\frac{\sqrt{d} a(\delta, \beta)}{\epsilon}\right)^{\frac{3\beta+2}{2\beta}}\right) \quad and \quad \mathcal{O}\left(\frac{\sqrt{d} a(\delta, \beta)}{\epsilon}\right),$$

*where $a(\delta, \beta) = (\log(1/\delta))^{\frac{\beta}{1+\beta}}$.*

*Remark* 3.7. First note that the oracle complexities under the weaker condition in Assumption 1.5 has an additional dimension factor $\sqrt{d}$. Hence, for a fixed value of $\delta$, for $\alpha = 1$ and $\beta = 1$ (i.e., finite variance case), we have the SFO complexity to be of order $\mathcal{O}(\epsilon^{-5/2})$ and $\mathcal{O}(d^{5/4}\epsilon^{-5/2})$ respectively. Furthermore, note that under our assumptions, only $(1 + \alpha)$ or $(1 + \beta)$ moment exists for the stochastic gradients. Consequently, as $\alpha \to 0$ or $\beta \to 0$, for a fixed value of $0 < \epsilon < 1$ and $\delta$, the SFO complexity tends to infinity, highlighting the difficulty of the problem.

## 3.3 ORACLE COMPLEXITY WITH CTBJ-BASED GRADIENT ESTIMATOR

In this section, we will use the mean estimator procedure proposed recently in [Cherapanamjeri et al., 2022], and

show that the dimension factor $\sqrt{d}$ appearing in the SFO complexity in part (b) of Theorem 3.6 could be removed, even under the weaker condition in Assumption 1.5.

**Theorem 3.8.** *Let Algorithm 2 be run with $\bar{G}_k$ as defined in (6), and with parameters $\alpha_k$, $\gamma_k$ and $\mu_k$ as defined in (8). If the stochastic gradients $G(x, \xi)$ satisfy Assumptions 1.5 and 3.1, and $m_k = \mathcal{O}(N^{\frac{2(\beta+1)}{\beta}})$, then with a target confidence $\delta > 2^{-\frac{m_k}{16000}}$ and $d \lesssim \log(1/\delta)$, we have*

$$\mathbb{P}\left(f(z_N) - f(x_*) \leq \frac{D_0 \max\left\{a(\delta, \beta), \frac{1}{N}a(\delta, \beta)^2\right\}}{N(N+1)}\right) \geq 1 - \delta,$$

*and the SFO and LMO complexity are respectively bounded by*

$$\mathcal{O}\left(\left(\frac{a(\delta, \beta)}{\epsilon}\right)^{\frac{3\beta+2}{2\beta}}\right) \quad and \quad \mathcal{O}\left(\frac{a(\delta, \beta)}{\epsilon}\right).$$

*Remark* 3.9. The proof of the above theorem is based on a concentration result for the gradient estimator (6), established in [Cherapanamjeri et al., 2022]. In comparison to part (b) of Theorem 3.6, the SFO complexity in Theorem 3.8 under Assumption 1.5 does not have the additional dimensional factor $\sqrt{d}$, thereby demonstrating the benefits of using the mean-estimation procedure proposed in [Cherapanamjeri et al., 2022]. However, this improvement does not come for free, as the per-iteration complexity of using the robust mean-estimator (6) is significantly higher than that of the clipped gradient based robust mean-estimator in (5), although it has a polynomial dependency on the problem parameters. See Section 5 for details regarding the per-iteration computational complexity of (6).

## 3.4 IMPROVING THE ORACLE COMPLEXITY

The above oracle complexity results based on clipped gradient based and CTBJ based robust gradient estimators have the following drawbacks. The $\epsilon$-dependency of the SFO complexity in part (a) of Theorem 3.6 under Assumption 1.4 or Theorem 3.8 under Assumption 1.5 is $\mathcal{O}(\epsilon^{-2.5})$ when $\alpha = 1$ and $\beta = 1$. In this section, we propose a bias-corrected clipped gradient based robust gradient estimation procedure with which SFO complexity of Algorithm 2 could be improved to $\mathcal{O}(d^2\epsilon^{-2})$ under an additional symmetry assumption on the distribution of the stochastic gradient. Hence, when high-accuracy solutions in low-dimensional settings are required, the bias-corrected clipped gradient based robust SFW algorithm could be preferred. We now introduce the symmetry assumption and the clipped gradient procedure.

**Assumption 3.10.** *Let the distribution of $G(x, \xi)$ be continuous and symmetric about $\mathbb{E}[G(x, \xi)] = \nabla f(x)$, for all $x \in \mathbb{R}^d$. Also, let the probability density function be a decreasing function with respect to $\|G(x, \xi) - \nabla f(x)\|$, for all $x \in \mathbb{R}^d$.*

**Proposition 3.11.** *For iteration $k$ the i.i.d. stochastic gradients $G(w_k, \xi_{k,j})$ are assumed to satisfy Assumptions 1.5 and 3.10. Let $\delta \in (0, 1)$. Consider the initial estimate defined as*

$$\widehat{G}_k := \underset{G(w_k, \xi_{k,j}): j \geq \frac{m_k}{2}}{\arg\min} \min\left\{r \geq 0 :$$

$$\sum_{\ell = \frac{m_k}{2}}^{m_k} \mathbb{1}\left\{\|G(w_k, \xi_{k,\ell}) - G(w_k, \xi_{k,j})\| \leq r\right\} \geq 0.3 m_k\right\},$$

$$(9)$$

*and consider the bias-corrected clipped gradient estimator $\bar{G}_k$ defined as*

$$\bar{G}_k := \frac{2}{m_k} \sum_{t=1}^{m_k/2} \min\left\{\frac{\left[\left(\frac{t}{\log(1/\delta)}\right)^{\frac{1}{1+\beta}} + 24\right]\sqrt{d}}{\|G(w_k, \xi_{k,t}) - \widehat{G}_k\|}, 1\right\} \times$$

$$\left[G(w_k, \xi_{k,t}) - \widehat{G}_k\right] + \widehat{G}_k. \quad (10)$$

*Then, as long as $m_k \geq 2\max\{50, 24^{1+\beta}\}\log(1/\delta)$, by recalling that $\bar{\Delta}_k = \bar{G}_k - \nabla f(w_k)$, we have*

$$\mathbb{E}[\bar{G}_k] = \nabla f(w_k) \quad and$$

$$\mathbb{P}\left\{\|\bar{\Delta}_k\| \leq 8\pi\sqrt{d}\left(\frac{\log(1/\delta)}{m_k}\right)^{\frac{\beta}{1+\beta}}\right\} \geq 1 - \delta.$$

*Remark* 3.12. The initial estimate defined in (9) is the same as that in the CTBJ estimator in (6). We show that this initial step, along with the clipped gradient procedure for a specific choice of clipping parameter (as defined in (10)) helps obtain an unbiased gradient estimator which is sufficiently concentrated.

We now leverage the result in Proposition 3.11 and show that one could obtain improved SFO complexity for certain ranges of $d$ and $\epsilon$ when running Algorithm 2 with the robust gradient estimator (10).

**Theorem 3.13.** *Let Algorithm 2 be run with $\bar{G}_k$ as defined in (10), and with parameters $\alpha_k$, $\gamma_k$ and $\mu_k$ as defined in (8).*

*(a) If the stochastic gradients $G(x, \xi)$ satisfy Assumptions 1.4, 3.1 and 3.10, and $m_k = \mathcal{O}(N^{\frac{3(\alpha+1)}{2\alpha}})$, we have for $(\log(1/\delta))^{\frac{1}{1+\alpha}} \geq [\Gamma(\frac{\alpha}{1+\alpha})\frac{1+\alpha}{\alpha}]^{\frac{1}{1-\alpha}}$,*

$$\mathbb{P}\left(f(z_N) - f(x_*) \geq \frac{C(d, \alpha, \delta)D_0\sigma}{N(N+1)}\right) \leq \delta, \quad (11)$$

*where*

$$C(d, \alpha, \delta) = \max\left\{\sqrt{d}a(\delta, \alpha)^{\frac{1}{\alpha}}, a(\delta, \alpha)^2\right\},$$

*and the SFO and LMO complexity are respectively bounded by*

$$\mathcal{O}\left(\left(\frac{C(d, \alpha, \delta)}{\epsilon}\right)^{\frac{5\alpha+3}{4\alpha}}\right) \quad and \quad \mathcal{O}\left(\frac{C(d, \alpha, \delta)}{\epsilon}\right).$$

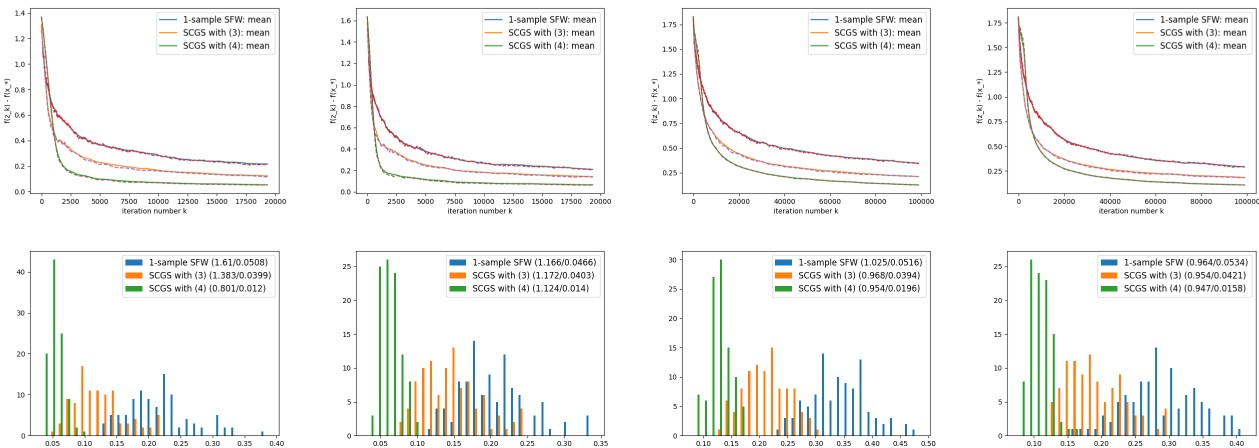

Figure 1: The two left and two right columns correspond to Pareto, Student-$t$ distributions with $d = 100$ and $d = 500$ respectively. **Top row:** Mean (solid lines) and Median (dotted lines) over 100 trails of iterations versus $f(z_N) - f(x_*)$ for $N = 100$. **Bottom row:** Histogram of $f(z_N) - f(x_*)$ for $N = 100$. Numbers in the legend correspond to *heavy-tail index/standard deviation.*

(b) *If the stochastic gradients $G(x, \xi)$ satisfy Assumptions 1.4, 3.1 and 3.10, and $m_k = \mathcal{O}(N^{\frac{3(\beta+1)}{2\beta}})$, the same conclusion in* (11) *holds with $\alpha$ replaced by $\beta$, with $C(d, \beta, \delta) := C(d, \alpha = \beta, \delta)$, for $(\log(1/\delta))^{\frac{1}{1+\beta}} \geq [\Gamma(\frac{\beta}{1+\beta})\frac{1+\beta}{\beta}]^{\frac{1}{1-\beta}}$.*

*Remark* 3.14. Note that when $\alpha = 1$, for any fixed value of $\delta$, $C(d, \alpha, \delta)$ is $\mathcal{O}(d)$. Hence, the SFO complexity is of order $\mathcal{O}(d^2\epsilon^{-2})$. Hence, when $d^2 < o(\epsilon^{-0.5})$ the SFO complexity of part (a) of Theorem 3.13 is better than part (a) of Theorem 3.6. A similar improvement holds for part (b). In Figure 3 (Appendix–Section 2), we compare Theorem 3.6, and Theorem 3.13 visually. For comparing part (a) of the respective theorems, we set $\alpha = 1$, $\epsilon = 10^{-10}$ and $\delta = 0.05$ vary $d$ from 200 to 1000 in steps of two hundred. For comparing part (b) of the respective theorems, we set $\alpha = 1$, $\epsilon = 10^{-6}$ and $\delta = 0.05$ vary $d$ from 2000 to 10000 in steps of two thousand.

## 4 CONSEQUENCES FOR PRACTICE

We now demonstrate the consequences of our theoretical results in practice. Among the robust gradient estimators in (5), (6) and (10), the *most practical one* (i.e., least per-iteration complexity) is the clipped gradient estimator in (5). Hence, we compare Algorithm 2 with the mini-batch average gradient in (4) and the clipped gradient estimator in (5) via experiments on sparse linear regression and multi-class logistic regression (presented in Appendix–Section 1 due to space limitations) with heavy-tailed covariates.

**Sparse Linear Regression:** We now provide simulation results for the regression problem described in (2). For our experiments, we consider the data vector $a \in \mathbb{R}^d$ to be a Pareto distribution with the exponent being 2.2 (which is asymptot-

ically a $t$-distribution with degrees of freedom 2.2). We ran Algorithm 2 with parameters as defined in (8) for 100 trials. Here, $L$ could be calculated analytically for our problem. For the choice of batch size, while Theorem 3.6 suggests $m_k = \mathcal{O}(N^{\frac{2(\alpha+1)}{\alpha}})$, we found that in our experiments setting $m_k = 500$ works well already. In Figure 1, we report the performance of Algorithm 2 with the clipped gradient estimator (5) and mini-batch average estimator (4). We also compare against the 1-sample SFW method from Mokhtari et al. [2020]. From the top row, we see that the clipped gradient estimator has faster convergence, i.e., it achieves higher accuracy with lesser iterations compared to the standard mini-batch averaging or the 1-sample SFW method. Furthermore, from the histogram in the bottom row, we see that the distribution of the last iterate has more fluctuations for the mini-batch average estimator and the 1-sample SFW method, compared to the clipped gradient estimator. We quantify this statement by reporting the *standard deviation* and also the *heavy-tailed index*, a widely used metric to quantify fluctuations [Hoaglin et al., 2000], which is defined as

$$\tau(F) = \frac{F^{-1}(0.95) - F^{-1}(0.5)}{F^{-1}(0.75) - F^{-1}(0.5)} \bigg/ \frac{\Phi^{-1}(0.95) - \Phi^{-1}(0.5)}{\Phi^{-1}(0.75) - \Phi^{-1}(0.5)},$$

where $\Phi$ is the distribution of a standard normal and $F$ is the empirical CDF obtained from the histogram. The results in Figure 1 **confirm our theoretical results and highlight the benefits of using robust versions of SFW algorithms for dealing with heavy-tailed data arising practice**.

**Summary and Outlook:** We proposed and analyzed robust versions of stochastic Frank-Wolfe type algorithms and established high-probability oracle complexity results. Our theoretical results are supported by numerical experiments on the problem of sparse linear regression and multi-class

logistic regression with heavy-tailed data. Developing oracle complexity results for robust projection-free algorithms under non-convexity, and developing more practical versions of robust projection-free algorithms are interesting problems that we plan to examine as future work.

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
