# OpenReview forum: "High-Probability Bounds for Robust Stochastic Frank-Wolfe Algorithm"
_auai.org/UAI/2022/Conference — UAI 2022 Oral_

### Official Review · Reviewer_ZVzv · 2022-04-02

**Q2(1) Originality/Novelty:** 3
**Q2(2) Significance/Impact:** 3
**Q2(3) Correctness/Technical Quality:** 3
**Q2(6) Clarity Of Writing:** 3
**Q6 Overall Score:** 7
**Q8 Confidence In Your Score:** 3

**Q1 Summary And Contributions:**

This paper provides high probability bounds for the stochastic Frank-Wolfe (SFW) optimization procedure. They analyze several settings: 1) SFW with standard gradient mean estimation when the stochastic gradients have sub-Gaussian tail 2)  SFW with clipped gradient estimators for heavy-tailed gradients 3) improvements to these results with better gradient mean estimation, and 4) SFW with bias correction in the gradient estimation.

**Q2 Assessment Of The Paper:**

More detailed information regarding each of these aspects is given below:

**Q2(4) Quality Of Experiments (Optional):**

3: Good: The experimental evaluation is adequate, and the results convincingly support the main claims.

**Q2(5) Reproducibility:**

3: Good: Key resources (e.g., proofs, code, data) are available and key details (e.g., proofs, experimental setup) are sufficiently well-described for competent researchers to confidently reproduce the main results.

**Q3 Main Strengths:**

The main strength of this paper is in its technical contributions -- there is strong technical progress on several fronts, namely, in producing high probability guarantees for SFW instead of guarantees in expectation, and also analyzing SFW with robust mean estimation for cases where the distribution is heavy-tailed. The latter is particularly significant and interesting because it is likely that many practical distributions are heavy-tailed.

**Q4 Main Weakness:**

The paper is somewhat dense in terms of mathematical formalisms and writing. Though this is understandable given that the paper is primarily theoretical and concerns a heavily mathematical topic, it does make the paper harder to digest. Towards making the paper easier to read, it might be helpful to add additional clarifying/high-level texts to explain lemma/theorem statements and implications. Furthermore, to highlight the technical contributions v.s. prior work, it would be nice to have a high level explanation of why this paper can achieve high probability bounds and prior work could not.

**Q5 Detailed Comments To The Authors:**

-- are there known lower bounds on SFO/LMO complexity for these settings?
-- how do the known results/contributions in the setting of this paper compare to other related settings in stochastic optimization? (e.g. methods for constrained problems which involve projection, unconstrained problems, etc.?) What are the challenges unique to the SFW setting?

**Q7 Justification For Your Score:**

This paper appears very solid on all fronts, with no major weaknesses -- the only suggestion I'd make to the authors is to try to make the presentation of the results less notationally dense and more accessible.

**Q9 Complying With Reviewing Instructions:**

1: Yes.

---

### Official Review · Reviewer_kEDW · 2022-04-15

**Q2(1) Originality/Novelty:** 3
**Q2(2) Significance/Impact:** 3
**Q2(3) Correctness/Technical Quality:** 3
**Q2(6) Clarity Of Writing:** 4
**Q6 Overall Score:** 7
**Q8 Confidence In Your Score:** 1

**Q1 Summary And Contributions:**

The paper considers stochastic Frank-Wolfe-type algorithms when the stochastic gradient is heavy-tailed. The paper proposes robust versions of the algorithm and strengthens the theoretical analysis of these algorithms by establishing high-probability bounds on oracle complexity.

**Q2 Assessment Of The Paper:**

More detailed information regarding each of these aspects is given below:

**Q2(4) Quality Of Experiments (Optional):**

3: Good: The experimental evaluation is adequate, and the results convincingly support the main claims.

**Q2(5) Reproducibility:**

3: Good: Key resources (e.g., proofs, code, data) are available and key details (e.g., proofs, experimental setup) are sufficiently well-described for competent researchers to confidently reproduce the main results.

**Q3 Main Strengths:**

1. The paper considers an issue of practical relevance and proposes a solution with careful theoretical analysis.
2. The high-probability results obtained strengthened the type results available in the literature (e.g., in expectation and requiring uniform variance bound).
3. The claims seem to be evidenced by numerical results.
4. The paper is very well-written.

**Q4 Main Weakness:**

(I do not have much to say in this regard).

**Q5 Detailed Comments To The Authors:**

Disclaimer: This paper does not fall into my area of expertise and my evaluation amounts to an educated guess. The paper was given to me urgently as one of its assigned reviewer had health issues.

I think this is a decent paper that makes a solid contribution to stochastic optimization algorithms, as I can imagine there are many use cases where the stochastic gradients are heavy tailed. In these cases, the standard algorithms, which are based on sample average of minibatch gradients, are inadequate. I like the fact that authors used estimators from robust statistics to overcome this problem. In particular, the one based on the clipped gradient estimator seems widely applicable. Further, the paper fills a gap in the theoretical analysis by providing novel high-probability bounds, which are more informative than the type of results previously available in the literature.



**Q7 Justification For Your Score:**

I think this paper (1) addresses a real issue for stochastic-gradient algorithms (2) proposes a valid solution and (3) provides a novel and tight theoretical analysis.

**Q9 Complying With Reviewing Instructions:**

1: Yes.

---

### Official Review · Reviewer_Yjus · 2022-04-18

**Q2(1) Originality/Novelty:** 3
**Q2(2) Significance/Impact:** 2
**Q2(3) Correctness/Technical Quality:** 3
**Q2(6) Clarity Of Writing:** 3
**Q6 Overall Score:** 7
**Q8 Confidence In Your Score:** 3

**Q1 Summary And Contributions:**

The paper considers stochastic conditional gradient algorithms for heavy-tailed stochastic gradients. It proposes to combine the stochastic conditional gradient sliding algorithms of Lam and Zhou with clipped-gradient estimates and a more sophisticated formulation from Cherapanamjeri et al. (2020). The paper proposes new concentration inequalities for martingales with heavy-tails to obtain its convergence results. Experiments confirm empirically faster convergence in the heavy-tail setting.

**Q2 Assessment Of The Paper:**

More detailed information regarding each of these aspects is given below:

**Q2(4) Quality Of Experiments (Optional):**

3: Good: The experimental evaluation is adequate, and the results convincingly support the main claims.

**Q2(5) Reproducibility:**

4: Excellent: Key resources (e.g., proofs, code, data) are available and key details (e.g., proof sketches, experimental setup) are comprehensively described for competent researchers to confidently and easily reproduce the main results.

**Q3 Main Strengths:**

- The paper consider several (two) robust gradient estimates  (although only a single one is considered in the experiments)
- The paper provides non-asymptotic convergence rates for $(\epsilon, \delta)$-optimality
- The paper leverage novel concentration results on martingales with heavy-tails to prove the convergence results
- The validation experiments are compelling

**Q4 Main Weakness:**

- The discussion of the related work in particular of the work using different robust gradient estimates in the context of stochastic gradient descent is too short. It would have been worth it to provide a more detailed discussion of different robust gradient estimates possible and of which proof techniques are transferable or not between the SGD setting and the a priori more complicated SCGS (especially since the considered algorithm is accelerated).

- The presentation of different algorithms can be improved in particular for the more complicated procedure presented in Appendix E. (Given that this procedure is only presented in the appendix, and seems to be too computationally heavy to be considered in the experiments it is not clear that it makes sense to include it in this paper.)

**Q5 Detailed Comments To The Authors:**

I found the paper quite interesting.
My main comment/ concern is the first comment below. The remaining point are less important, and are mainly suggestions to potentially improve the writing of the paper.

1. The review of the related work is too short and is very close to being limited to a list of references. The sentences that introduce these references provide extremely loose characterization of what these papers are about. The paragraph concludes by saying that the papers listed are not about projection-free robust optimization, which is somewhat disappointing, because the author give the impression of turning their back on the prior literature, and because the really interesting question is whether the current work can/did get inspired by ideas proposed in this existing literature.
In particular, in the cited papers on stochastic gradient descent with heavy-tailed input data distribution (e.g [40]) or heavy-tailed noise distribution (e.g. [17]), both similar gradient clipping techniques but also other gradient approximation techniques such as the median of means are considered. Moreover relevant concentration results are considered. I am therefore disappointed that the paper does not discuss more these results, the ones that could potentially be used in the context of the conditional gradient algorithm and the ones that could not (?), and why specific proof techniques are needed for the conditional gradient case. After all, once some control over the concentration of the gradient estimate is obtained, it could seem a priori that they could be similarly used in the convergence proof schemes of different algorithms. It would be useful to explain why specific results are needed.

2. The proof of Lemma 3.1 it the submitted paper seems to be following essentially the proof of Theorem 3.1 in [31] with very minor differences between the statement of Lemma 3.1 in this paper and the equation (3.8) on page 14 in [31]. If the Lemma is essentially the same, the paper should acknowledge this very explicitly, by saying that the authors use an intermediate result of the proof of Theorem 3.1 in [31].

3. The paper says that it will consider the stochastic conditional gradient sliding of Lan and Zhou, however the fact that the text then explains that the paper will first introduce a subroutine and comments on the meaning of the first part plus the fact that the algorithm is rewritten in two parts with different notations with new names for the two algorithms confused me and it took me some time to realize that it was indeed exactly the algorithm of Lan and Zhou in which only the calculation of $\bar{G}_k.$
I would encourage the authors to be more clear about this and to state explicitly at the beginning that this will be the same algorithm (stochastic conditional gradient sliding) in which the average stochastic gradient will be replaced by a robust estimate of the gradient, and to say explicitly that the algorithm is rewritten as 2 algorithms.

4. In Assumption 1.3, the fact that the right hand side of the inequality is 1 is surprising: why not $\sigma^{1+\beta}$, as in Assumption 1.2?

5. In equation (5), I would suggest naming the event $A_{j}$ because it is different for each $j$. Is the normalization really $m_k$? And not the number of events that occur?
In other words, in equation (5), can you confirm that what is computed is the mean of these gradients that are not too large but really a mean over $m_k$ terms with some of these terms set to $zero$. If a large number of gradients are discarded the gradient will be even more serious biased towards $0$, no?


6. In Algorithm 8, what are the dimensions of the matrix $X$? Is the optimization over the set of integers $b_i$ as well? Or is $b_i$ fixed? Same question for the $v_j$s?
Also the fact that $v_{b_i}$ is a vector while $v_j$ is an integer is confusing.
Having an informal explanation about what the semi-definite optimization problem solves would be useful.

7. The presentation of the algorithm of [9] presented in Appendix E is unfortunately not sufficiently clear and self-contained to be understandable. The reader is forced to go back to [9] anyway.

8. Minor comments on notations, writing, typos, errors, etc:

- What is $\alpha$ in Algorithm 5? The same as in Assumption 1.2?

- In section 4, in the title and in the first sentences: "for practice" would be better formulated as "in practice".

- The text has a lot of small grammar mistakes, e.g. the verb are not properly conjugated, and should be in the form of the past participle instead of the infinitive or in the infinitive instead of the gerund.

-  In Algorithm 1, $\xi$ should be $x$. In Algorithm 2, $z_0$ is not initialized.

- Also, formally there is a bug in Algorithm 1, because if $\kappa=1$ then $\bar{y}_t$ is not computed and so when the algorithm exits the "while" loop $\bar{y}_t$ is undefined, so it is not possible to output it...

- I would suggest to introduce the computation of the robust stochastic before the Algorithms 1 and 2 are introduced. The paper will not have to cite equations that are in the following section...


**Q7 Justification For Your Score:**

- The paper is compelling and interesting, and makes a clear contribution for a fairly specific setting.

**Q9 Complying With Reviewing Instructions:**

1: Yes.

---

### Official Review · Reviewer_fNi8 · 2022-04-19

**Q2(1) Originality/Novelty:** 2
**Q2(2) Significance/Impact:** 2
**Q2(3) Correctness/Technical Quality:** 3
**Q2(6) Clarity Of Writing:** 4
**Q6 Overall Score:** 5
**Q8 Confidence In Your Score:** 4

**Q1 Summary And Contributions:**

The paper develops regularity conditions for heavy-tailed distributions, such that stochastic FW-type optimisation methods can be given rigorous convergence guarantees. This aims to improve our understanding of the stochastic behaviour of these optimisations, as the literature usually considers more idealised distribution classes. Technically, this boils down to introducing robust estimators in place of averaging gradients and studying their convergence under various conditions.

**Q2 Assessment Of The Paper:**

More detailed information regarding each of these aspects is given below:

**Q2(4) Quality Of Experiments (Optional):**

3: Good: The experimental evaluation is adequate, and the results convincingly support the main claims.

**Q2(5) Reproducibility:**

2: Fair: Key resources (e.g., proofs, code, data) are unavailable but key details (e.g., proof sketches, experimental setup) are sufficiently well-described for an expert to confidently reproduce the main results.

**Q3 Main Strengths:**

- very good motivation: since FW-type methods are sensitive to stochasticity in gradient computation and since real-world distributions are often heavy-tailed, what are regularity conditions for such distributions?
- quantitate approach: the authors develop their analysis using established probabilistic tools: moment conditions, MGFs etc.



**Q4 Main Weakness:**

I think that the auxiliary technical results do not utilise/credit the relevant literature to the full extent.
As a result, despite the heavy content, the paper fail to bring the expected technical novelty. More precisely:
- the properties of the proposed regularity/moment conditions have been discussed already in the literature (relevant to proofs in section C)
- the robust mean estimators of similarly-defined heavy-tailed distributions have been also discussed in the literature (relevant to invoking Assumptions in section D)

In addition, I think the paper would benefit from expanding the empirical section and providing full details on the experiments.

**Q5 Detailed Comments To The Authors:**

- Playing with the Assumptions and their properties (particularly part C of the appendix) seems to replicate to a large extent what we already know about Sub-Weibull distributions, please consult and compare with https://hal.inria.fr/hal-02545121v2/file/paper_2020_Sub_Weibull__arXiv_.pdf and similar papers. Another relevant references may be those on a-stable distributions.

- Technical results from the literature on robust estimation have not been fully exploited. See for example the discussion of trimmed-mean or median-of-means estimators  in "Bandits With Heavy Tail" or "Mean estimation: median-of-means tournaments", under same moment conditions as those used in the current paper.

- On empirical results, I think it would be great to provide the reader with a larger scale experiment, also including actual vs expected charts.

- No clarity on the usage of Pareto Distribution, it is confusingly/wrongly referred to: p.8 "For our experiments, we consider [...] a t-distribution or Pareto distribution with degrees of freedom 2.2." (Pareto should have an exponent parameter specified)

- The technical content is bit hard to read. I would suggest more asymptotic notation (use O(N^2) in place of N(N+1) etc) as well as the convention on probabilistic errors from the robust estimation literature (so you explain 1-delta probability in words, cf ROBUST MULTIVARIATE MEAN ESTIMATION: THE OPTIMALITY OF TRIMMED MEAN)



**Q7 Justification For Your Score:**

The motivation and idea on their own are great. Nevertheless, I believe the paper should do better on
a) utilising the relevant literature and clearly benchmark its own development, with regards to concentration inequalities and robust estimation; I would love to see a better explanation why some results from the literature are not applicable / not comparable and how the current paper contributes.
b) (possibly) expanding the empirical section.

**Q9 Complying With Reviewing Instructions:**

1: Yes.

---

### Decision · Program_Chairs · 2022-05-15

**Decision:**

Accept (Oral)

**Comment:**

Meta Review: The reviewers are happy with the authors’ responses. The reviewers suggest the following things: (i) The authors should add the lack of discussion of the related literature; and (ii) the authors should better modularize and differentiate what is credited and established vs developed and new. Please also add the discussions with the reviewers into the final version.